# A Cohort Study on the Effect of Parental Mind-Mindedness in Parent−Child Interaction Therapy

**DOI:** 10.3390/ijerph19084533

**Published:** 2022-04-09

**Authors:** Merlijn Meynen, Cristina Colonnesi, Mariëlle E. Abrahamse, Irma Hein, Geert-Jan J. M. Stams, Ramón J. L. L. Lindauer

**Affiliations:** 1Levvel, Academic Center for Child and Adolescent Psychiatry, Meibergdreef 5, 1105 AZ Amsterdam, The Netherlands; m.abrahamse@levvel.nl (M.E.A.); i.hein@levvel.nl (I.H.); r.lindauer@levvel.nl (R.J.L.L.L.); 2Department of Child and Adolescent Psychiatry, Amsterdam UMC, Location Academic Medical Center, University of Amsterdam, Meibergdreef 9, 1105 AZ Amsterdam, The Netherlands; 3Research Institute of Child Development and Education, University of Amsterdam, Nieuwe Achtergracht 127, 1001 NG Amsterdam, The Netherlands; c.colonnesi@uva.nl (C.C.); g.j.j.m.stams@uva.nl (G.-J.J.M.S.)

**Keywords:** PCIT, mind-mindedness, disruptive behavioral problems, parenting skills

## Abstract

Parent−child interaction therapy (PCIT) is a short-term, evidence-based intervention for caregivers with children aged between 2 and 7 who exhibit behavioral problems. PCIT is effective, but has a high attrition rate ranging from 27% to 69%. We hypothesize that a low level of parental mind-mindedness—the parent’s propensity to treat the child as an intentional agent with its own thoughts and emotions—might contribute to premature attrition or cause families to profit less from treatment. To test these hypotheses, we performed a retrospective cohort study in a time-limited, home-based PCIT sample (*n* = 19) and in a clinic-based PCIT sample (*n* = 25), to investigate whether parents with a medium-high level of mind-mindedness differ from parents with a medium-low level of mind-mindedness in the outcome measures of PCIT (child’s behavioral problems, parenting skills and stress and mothers’ anxious and depressed symptoms). Furthermore, we examined if mind-mindedness was related to attrition and (for clinic-based PCIT only) number of sessions. Repeated measures ANOVA showed that mothers with a medium-high level of mind-mindedness displayed more improvement in two parenting skills benefiting a positive parent−child interaction. Furthermore, we found a group effect of mind-mindedness in the PCIT-home sample, with mothers with a medium-high level of mind-mindedness showing better results on most outcome measures. Our findings suggest that adding a mind-mindedness improving intervention prior to or during PCIT could benefit mothers with a medium to low level of mind-mindedness.

## 1. Introduction

Disruptive behavioral disorders are highly prevalent among preschool children [1], and they are one of the most common reasons for referral to mental health services at this age [2]. If untreated, behavioral problems have a high degree of persistence [3], and increase the risk of developing mental health problems later in life such as anxiety disorders, depression, antisocial personality disorders, and addiction [4]. Other long-term outcomes include academic difficulties [5], unemployment and family problems [6], as well as criminal and violent behavior into adolescence and adulthood [7]. These outcomes result in high costs for mental health, law enforcement, educational and social services, and add to the necessity of effective, evidence-based interventions for behavioral problems at an early stage in the child’s development.

Parent training programs, which use the parents as the primary agent of change, have proven to be the most effective method for treating young children with behavioral problems [8]. A widely used and evidence-based parent training program is parent−child interaction therapy (PCIT) [9], originally developed for children aged between two and seven years. The treatment is based on both attachment [10] and social learning [11] theory, and is aimed at improving the quality of the parent−child relationship and teaching the parent effective parenting skills to better manage their child’s behavior [12]. Multiple studies have shown medium to large effect sizes of PCIT in reducing the child’s disruptive behavior and parenting stress and improving parenting skills [13], with growing evidence that PCIT prevents against child maltreatment [14,15,16].

Despite its effectiveness, premature attrition from treatment remains a serious problem for PCIT, with attrition rates ranging from 27% to 69% [17,18,19,20]. Largely, these high attrition rates are a result of barriers most families referred to community mental health services face, due to low socioeconomic resources [21]. These barriers to treatment are often very pragmatic, and include limitations in transportation, provision of childcare, and disruptions in work [22,23]. Furthermore, other problems such as financial or parental mental health problems might complicate staying motivated for treatment [24]. To increase the accessibility and effectiveness of PCIT for families with low resources, a home-based adaptation of PCIT has been developed [25]. The core principles of treatment of PCIT stay intact, but the home-based adaptation provides several key advantages compared to the traditional clinic-based PCIT: it eliminates the logistical barriers, it has strong ecological value because families are observed in their natural setting, while learned skills can be generalized faster to other situations and contexts [21,26]. Studies on home-based PCIT have shown similar outcomes compared to clinic-based PCIT, so the advantage is mostly found in reaching families otherwise difficult to reach. However, the results on lower rates of attrition for PCIT-Home compared to PCIT-Clinic have been inconclusive [20,21,27,28,29,30,31]. Furthermore, PCIT has been adapted to a time-limited version, based on the less-is-more principle of Bakermans−Kranenburg [32], which assumes that brief interventions with a moderate number of sessions and a clear end point are more effective than interventions with a large number of sessions. In this line, Abrahamse et al. [33] examined the effectiveness of a time-limited, home-based adaptation of PCIT, with lower rates of attrition (15%), and a potentially protective effect against child maltreatment.

Home-based and time-limited adaptations of PCIT point to the importance of understanding the mechanisms and effective elements behind PCIT so it can be adjusted to the needs of specific families to make it more effective and prevent attrition. An important aspect of PCIT and its possible effect on treatment outcomes and attrition, which has not yet been investigated, is parental mentalization [34]—the parent’s ability to represent and hold in mind the internal states of their child. PCIT acknowledges the importance of enhancing the parent−child relationship through positive changes enacted by the parent that support a sense of security in the child. This demands of the parent to become more attuned to their child’s psychological needs and responding to such signals in a predictable, consistent, and sensitive manner [35]. Therefore, while not specifically highlighted in PCIT, the parental ability to take the perspective of the child or ‘to mentalize about the child’ might be a key prerequisite for the positive change in the parent’s behavior and for the efficacy of PCIT. In other words, we expect that parents who are less able to understand and take the perspective of their child are less able to change their behavior accordingly, and therefore will profit less from treatment or are at risk of premature attrition. To test these hypotheses, we performed a retrospective cohort study, investigating if parental mind-mindedness, which is a form of parental mentalization, affects the outcome of PCIT.

Mind-mindedness [36] refers to the parents’ tendency to consider and to treat their children as individuals with an independent mind, rather than as entities with needs that must be met [37,38]. It can be measured by observing the parent−child interaction in a ‘free play session’ and coding systematically how often the parent makes a mind-related comment about the child (e.g., the child’s emotions, desires, intentions, thoughts) [39]. Research shows that mind-mindedness is associated with the child’s development of social understanding, parent−child secure attachment (for a review, see [40,41]), language development, and school readiness [42,43]. There is also emerging evidence that mind-mindedness protects against the development of child behavior problems [44,45,46,47,48]. Studies investigating parenting programs aimed at increasing mind-mindedness of parents with adopted or foster children showed improvements in children’s behavioral problems, attachment, and parenting stress [49,50,51]. Mind-mindedness has not been measured or investigated in PCIT before.

### The Present Study

The aim of the present study was to investigate to what extent a caregiver’s use of mind-mindedness measured before the start of treatment has an effect on the outcome of both parent−child interaction therapy provided in a time-limited, home-based setting (PCIT-Home) and in a clinic-based setting (PCIT-Clinic). We expected a caregiver’s use of mind-mindedness would have a moderating effect on the outcome measures, resulting in more decrease of the child’s behavioral problems, parenting stress, and mothers’ anxious and depressive symptoms, and more improvement in parenting skills. We also expected a group effect of mind-mindedness, with mothers with a medium-high level of mind-mindedness reporting fewer behavioral problems in their children, experiencing less parenting stress and anxious and depressive symptoms, and showing better parenting skills. Furthermore, we investigated whether mind-mindedness is associated with a lower risk of premature attrition and—for the PCIT-Clinic sample only—fewer sessions.

## 2. Materials and Methods

### 2.1. Participants

Two different intervention groups were included in the study: a time-limited, home-based PCIT sample (PCIT-Home; *n* = 19) and a clinic-based PCIT sample (PCIT-Clinic; *n* = 25). Table 1. presents the main characteristics of both groups. We analyzed the PCIT-Home and PCIT-Clinic groups separately, because we expected different outcomes due to significant differences in study design (PCIT-Clinic consisted of an immediate treatment group, PCIT-Home also had a waitlist control group that started treatment after 2 months; different inclusion and exclusion criteria; different follow-up periods), sample characteristics and treatment procedure. The most important difference in sample characteristics was the relation of the mother to the child: the PCIT-Home sample contained 31.6% foster mothers, while the PCIT-Clinic sample contained only biological mothers.

In both samples, approximately 40% of the families were single mothers and 13% of the single mothers were divorced. Thirty-five percent (PCIT-Home) and 73% (PCIT-Clinic) of the non-single mothers were married. The majority of the mothers were employed (PCIT-Home 74%; PCIT-Clinic 64%) and a large percentage attended only lower education (PCIT-Home 44.4%; PCIT-Clinic 52.0%). The most important difference between the groups was the relationship of the mother to the child: the PCIT-Home sample contained 31.6% foster mothers, while the PCIT-Clinic sample contained only biological mothers. The large majority of the children were boys with a mean age of around 5 years. Based on the classification criteria for ethnic background of Statistics Netherlands [52], in the PCIT-Home sample 58% of the children had a Dutch ethnicity, 37% had a non-Western migration background, and 5% a Western migration background. In the PCIT-Clinic sample, 80% of the children had a Dutch ethnicity, 16% had a non-Western migration background, and 4% a Western migration background. Child maltreatment history was based on the criteria of Barnett et al. [53] using the Maltreatment Classification System.

#### 2.1.1. PCIT-Home Sample

The PCIT-Home sample consisted of 19 children participating in a longitudinal study on the prevention of child maltreatment [33]. Children were referred for treatment of disruptive behavioral problems and a history of maltreatment of the child through the usual community channels (mental health services, internal referrals, child’s school or by general practitioners) and as with the PCIT-Clinic sample, treatment took place in a community mental health center in the Amsterdam region, the Netherlands. Recruitment for study participation took place from October 2014 to June 2016, and the data collection continued until December 2016.

Inclusion criteria for the original study were: (1) disruptive behavioral problems or risk of child maltreatment were reason for referral, (2) children between two and seven years. Serious concerns about a child’s safety in the home situation were an exclusion criterion.

The original sample from Abrahamse et al. [33] consisted of 20 children. One family was excluded from the present study because parents did not give permission for scientific use of the video-recordings (and therefore mind-mindedness could not be assessed).

#### 2.1.2. PCIT-Clinic Sample

The PCIT-Clinic sample consisted of 25 children and their parents participating in a previous longitudinal study on the risk factors for attrition from PCIT [17]. Children were referred for treatment of disruptive behavioral problems through the usual community channels (child protection services, mental health services, internal referrals or by general practitioners), and treatment took place in a community mental health center in the Amsterdam region, the Netherlands. Recruitment for study participation took place from June 2009 to December 2012.

Inclusion criteria for the original study were: (1) disruptive behavioral problems were a reason for referral, (2) children between two and seven years, and (3) the parents were Dutch- or English-speaking. Exclusion criteria were children’s clinical signs of developmental or physical disabilities, parental learning disabilities (IQ < 80), parental substance use disorders, and serious concerns about a child’s safety in the home situation.

The original sample consisted of 40 children. For the present study, three families were excluded because video-observations of the parent−child interaction could not be retrieved. Furthermore, 12 dropouts were excluded because they contained large amounts of missing data (missing values dropouts (*n* = 12): 32.47%; missing values completers (*n* = 25): 11.93%). The completers of the present study (*n* = 25) contained four families which were originally coded as dropouts, but were included as completers because treatment was considered mostly successful by both the parents and the therapist even though they did not fully meet the criteria for a successful treatment according to protocol (i.e., the child’s disruptive behavior was brought within normal limits and the caregiver reached mastery criteria for the CDI and PDI skills [54].

### 2.2. Time-Limited, Home-Based PCIT and Clinic-Based PCIT: Similarities and Differences

Time-limited, home-based (PCIT-Home) and clinic-based parent−child interaction therapy (PCIT-Clinic) both consist of two treatment phases, child-directed interaction (CDI) and parent-directed interaction (PDI). The first phase focuses on improving the parent−child relationship and the second phase on child compliance where the parent takes a more leading role. Both treatment phases start with a didactic session followed by weekly coaching sessions where the parent is coached by the therapist during play sessions with their child. Parents participating in PCIT-Clinic receive feedback on their parenting skills behind a one-way mirror via an in-ear earbud. Parents move to the PDI phase only after mastering the CDI-criteria (10 behavioral descriptions, 10 reflections, 10 labeled praises, and fewer than three commands, questions, or negative verbalizations during a 5-min observation). The PDI phase continues until parents reach the mastery criteria for the PDI skills (75% effective commands, 75% correct follow-through (praise or warning); if time-out, correct follow through) [54] and rate their child’s behavior within normal limits (Eyberg Child Behavior Inventory Intensity Scale ≤ 114). As a consequence, the number of total sessions may vary among families participating in PCIT-Clinic. In contrast, PCIT-Home consists of four CDI sessions and four PDI sessions [33], regardless of mastery skills or the child’s behavior at the end of treatment. Additionally, treatment takes place at the parent’s home and because there is no one-way screen available, the therapist coaches the parents from a distance using a wireless headset or the therapist takes a strategic position behind the parent.

### 2.3. Procedure

Baseline assessment was conducted prior to the start of the intervention, and post-treatment assessment was carried out immediately after the researcher was informed about treatment completion or termination. Follow-up assessments were performed six months after the post-treatment assessment for PCIT-Clinic. The follow-up of the PCIT-Home sample was performed after 2 months because of a different study design: besides an immediate treatment group there was a waitlist control group, which started treatment after 2 months, while the follow-up period was evened up with the waitlist period. Ethical approval for the study was obtained from the Medical Ethics Committee of the Academic Medical Center of Amsterdam, approval codes 2014.252 (PCIT-Home sample) and METC2009_081 (PCIT-Clinic sample).

## 3. Measures

Summaries of used measures are listed below. The adult self-report (ASR) method was available in the PCIT-Clinic sample only. Parenting stress was assessed using the Dutch Parenting Stress Questionnaire (OBVL) for PCIT-Home and the Parenting Stress Index Short Form (PSI-SF) for PCIT-Clinic. All assessments were completed by the female caregivers only.

### 3.1. Mind-Mindedness

We assessed mind-mindedness using video-recorded parent−child interactions in the home setting before the start of the PCIT treatment. These recordings consisted of three situations, with each situation requiring an increasing degree of parental control and direction [55]. We chose to code mind-mindedness during child-led play (CLP) only, since this situation reflects the ‘free play session’ from Meins and Fernyhough’s guidelines [39] for coding mind-mindedness. During CLP, parents received a written instruction to let the child choose the toys or game of their preference and the parent is to follow and play together with the child. Five minutes of this parent−child interaction was recorded and transcribed. This article’s first author then coded these transcripts for mind-mindedness following the manual of Meins and Fernyhough [39]. Parents’ comments were classified as either directed at the child’s mental state or not, with mind-related comments referring to the child’s desires/preferences (e.g., “do you like this color?”), cognitions (e.g., “do you know what this means?”), emotions (“you are sad”), epistemic states (“you are joking!”) or comments that were clearly dialogue intended to be spoken by the infant (e.g., the mother says: “can you please help me, Mom?”). Each mind-related comment was then coded as appropriate (the coder agreed with the parent’s reading of the child’s mental state) or non-attuned (the coder believed the parent’s reading of the child’s inner state was incorrect or not (enough) related to the infant’s current activity). Percentages of mind-mindedness were calculated ((*n* of appropriate or non-attuned mind-related comments/total number of comments) × 100) [39] and used in the analyses. Inter-rater reliability was assessed on 6 out of 19 transcripts for PCIT-Home (32%) and 9 out of 37 transcripts for PCIT-Clinic (24%). Intra-class correlations coefficients (ICC; two-way random effects model with an absolute agreement definition) on the number of appropriate mind-related comments, were 0.98 for PCIT-Home and 0.82 for PCIT-Clinic. According to the guideline of Koo and Li [56], ICC interrater agreement measures between 0.75 and 0.90 can be interpreted as good, and above 0.90 as excellent. Disagreements were resolved through discussion. All mind-related comments were scored as appropriate, meaning we did not detect any use of non-attuned comments. This finding is in line with other studies in which mind-mindedness was investigated beyond infancy [57].

### 3.2. Dyadic Parent–Child Interaction Coding System

The Dyadic Parent−Child Interaction Coding System (DPICS) [55] is a behavioral observational coding system that was used to measure the quality of parent−child interactions during three 5-min situations. In the first situation, child-led play (CLP), the parent is instructed to follow the play chosen by the child. In the second situation, parent-led play (PLP), the parent is instructed to choose a game of their liking and the child has to comply with the parent’s rules. In the last situation, clean-up (CU), the parent is instructed to order their child to clean up the toys without helping the child. For the present study, two child categories and four parent categories commonly used in research [55,58,59] were included in our analyses. The child categories were inappropriate behavior (including negative talk, negative touch, yell and whine; coded in all three situations) and percentage of compliance (coded in PLP and CU only). The four parent categories were the percentage of positive following (including behavior descriptions, reflections, labeled praise, and unlabeled praise; coded in CLP only); the percentage of negative leading (including commands, questions, and negative talk; coded in CLP only); praise (the sum of all praise in the three situations, including labeled and unlabeled praise); and demandingness (the sum total of indirect and direct commands, coded in PLP and CU only). The observations were recorded and both samples were coded by independent master-level research assistants and undergraduate students and had 80% agreement with the first author of the original studies [17,33]. All observations were transcribed and one random situation (CLP, PLP, or CU) per child was also coded by a second coder to estimate inter-rater reliability. The overall percentage agreement across the used DPICS categories was 0.91 (average Kappa; range 0.78–0.98) for the PCIT-Clinic sample and 85% (range 81–98%) for the PCIT-Home sample.

### 3.3. Eyberg Child Behavior Inventory

The Eyberg Child Behavior Inventory (ECBI) [60] is a parent rating scale used in both clinical practice and research settings to assess child disruptive behavior in children aged between 2 and 16 years. The ECBI has 36 items and measures the frequency of child behavioral problems (intensity scale) ranging from one (*never*) to seven (*always*) and to what extent the parents experience these behaviors as problematic (problem scale) on a dichotomous scale (*yes* or *no*). For the intensity scale, scores range from 36 to 252 and for the problem scale, scores range from 0 to 36. The published cutoff scores for the intensity scale are ≤132 and ≤15 for the problem scale. The ECBI is translated into Dutch with well-established reliability, with high internal consistencies (Cronbach’s alpha) for both the intensity scale (*α* = 0.93) and the problem scale (*α* = 0.91) [61,62]. In the present study, we also found high internal consistencies for both the PCIT-Home sample (intensity scale *α* = 0.94; problem scale *α* = 0.92) and the PCIT-Clinic sample (intensity scale *α* = 0.91; problem scale *α* = 0.89.

### 3.4. Strengths and Difficulties Questionnaire

The Strengths and Difficulties Questionnaire (SDQ) [63] is a brief 25-item questionnaire for parents with children between 3 and 16 years of age, which assesses the child’s emotional and behavioral problems. The SDQ has five subscales; for this study only the subscales of conduct problems and hyperactivity/inattention problems were included. Every subscale contains the sum of five items with three response categories (0 = *not true*, 1 = *somewhat true* and 2 = *certainly true*). The internal consistency (Cronbach’s alpha) in the validation study of the SDQ for the subscale conduct problems was 0.63 and for the subscale hyperactivity/inattention problems 0.77 [64]. In the present study, the internal consistencies were 0.47 (conduct problems) and 0.63 (hyperactivity/inattention problems) for PCIT-Home and 0.56 (conduct problems) and 0.83 (hyperactivity/inattention problems) for PCIT-Clinic.

### 3.5. Adult Self-Report

To measure internalizing psychopathology in the mothers, the adult self-report (ASR) method [65] was used. The ASR is a 123-item self-report questionnaire and includes eight empirically based syndrome scales: ‘rule-breaking behavior’, ‘aggressive behavior’, ‘intrusive’, ‘withdrawn’, ‘somatic complaints’, ‘anxious/depressed’, ‘thought problems’, and ‘attention problems’. For this study we only included the scale ‘anxious/depressed’ because it was found to be a predictor for attrition in previous research [17]. The internal consistency (Cronbach’s alpha) of the scale ‘anxious/depressed’ in the validation study of the ASR was 0.88 [65]. In the present study, the internal consistency was 0.93.

### 3.6. Parenting Stress Index Short Form & Dutch Parenting Stress Questionnaire 

To measure parenting stress, the Dutch Parenting Stress Questionnaire (Opvoedingsbelasting vragenlijst, OBVL) [66] was used for the PCIT-Home sample and the Dutch translation of the Parenting Stress Index Short Form (PSI-SF) [67] for the PCIT-Clinic sample. The OBVL has 34 items that measure problems within the parent–child relationship, parenting problems, depressive moods, role restriction, and health problems. The overall parenting stress scale was calculated by the *T*-score of the sum of all items, ranging from 34 to 136. A sum score above 59 indicates a clinical level of parenting stress. The internal consistency (Cronbach’s alpha) in the validation study of the OBVL was 0.89–0.91 [66]. In the present study, the internal consistency was 0.70. The PSI-SF has 25 items that measure dysfunctional parent−child interaction, stress in the parent−child relationship, and difficult behavior of the child with a 6-point scale ranging from 1 (*completely disagree*) to 6 (*completely agree*). The sum of all items was used as the overall parenting stress scale. According to published norms [68], a sum score above 74 indicates a clinical level of parenting stress. The internal consistency in the validation study of the PSI-SF was 0.91 [67] In the present study, the internal consistency was 0.94.

## 4. Statistical Approach

The data were analyzed using IBM SPSS Statistics, version 26 (IBM Corp., Armonk, NY, USA). Both datasets contained missing data: PCIT-Home 21.1% and PCIT-Clinic 11.9% (total missing values). The missing values were completely at random, with Little’s MCAR-test for PCIT-Home χ^2^ (24) = 23.63, *p* = 0.483 and for PCIT-Clinic χ^2^ (82) = 84.31, *p* = 0.409. Therefore, conditions were met to impute the missing data. We performed multiple imputation (MI), a method which has proven appropriate with small sample sizes and large proportions of missing data of up to 50% [69]. Using MI, we created 5 copies of the original datasets, and missing values of the outcome variables in each dataset were replaced with independent random draws from the predictive distribution of the missing values under a specific model (the imputation model). These multiple results were combined into one dataset (one for PCIT-Home and one for PCIT-Clinic), which we used for our analyses. For the analyses of moderating effect of mind-mindedness on the different treatment outcomes, series of repeated measures analysis of variance (ANOVA) were conducted. For these analyses, the percentage of appropriate mind-mindedness was divided into two groups (MM-*low* and MM-*high*) by median. Post hoc analyses were conducted by means of adjusted SIDAK comparisons and alpha levels were adjusted using Tukey in the correlations. We performed additional sensitivity analyses with covariates to control for confounding. The effect sizes for the repeated measures ANOVA were presented in terms of partial eta squared (*η_p_*^2^: 0.01 = *small*, 0.06 = *medium*, 0.14 = *large*).

Spearman nonparametric correlations were computed to test the relation between mind-mindedness and dropouts and (for PCIT-Clinic only) total sessions.

Assumptions relevant to all analyses were verified. The variables ECBI Intensity, ECBI Problem, OBVL, PSI-SF, DPICS Non-Compliance, DPICS Positive Following, and DPICS Negative Leading were normally distributed, with values of skewness/SE_skew_ and kurtosis/SE_kurt_ between −1.96 and +1.96 [70]. Outlier scores (i.e., Z-scores exceeding ± 3) were replaced with scores one unit larger than the next most extreme score [71].We found two variable outliers for PCIT-Home (DPICS Total Praise T1; DPICS Demandingness T3) and two for PCIT-Clinic (DPICS Inappropriate Child Behavior T2; ASR T3). The outliers were winsorized, after which the data were normally distributed. We applied winsorization, as this method allows weight modification without discarding the values of outliers, does not cause a bias based on under- or overestimation (e.g., trimming) and because the nature of the population distributions was not known (e.g., robust estimation method) [72]. No multivariate outliers were found using Mahalanobis distance.

To test whether our samples had enough power to discern significant change between pre-, post-, and follow-up tests, we performed a power analysis using the program G*Power [73]. We calculated power separately for PCIT-Home and PCIT-Clinic. Because we expected different correlations between the observational instrument (DPICS) and the questionnaires (ECBI, OBVL, PSI-SF, ASR), we also conducted separate calculations for these instruments. The power analysis was conducted for a repeated measures ANOVA with three repeated measures. We assumed an effect size of 0.30, an alpha of 0.05, and a power of 0.80. For the PCIT-Home sample, the mean correlation of the observational instrument was 0.30 between the pre-, post-, and follow-up tests, and for the questionnaires the mean correlation was 0.65. The power analyses indicated that a sample size of 20 children for the observational instrument, and a sample size of 12 children for the questionnaires, would be adequate to detect a significant difference between pre-, post-, and follow-up tests. For the PCIT-Clinic sample, the mean correlation of the observational instrument was 0.25 between the pre-, post-, and follow-up tests, and for the questionnaires the mean correlation was 0.50. Respectively, sample sizes of 22 children and of 16 children would be adequate to detect a significant difference between pre-, post-, and follow-up tests.

## 5. Results

### 5.1. Preliminary Analyses

We investigated whether mind-mindedness was correlated with the outcome variables at T1. For PCIT-Home, mind-mindedness was only significantly positively correlated with the parental DPICS subcategory ‘positive following’, *r*(18) = 0.53, *p* = 0.019. Mind-mindedness was not significantly related to all other outcome variables (ECBI Intensity, ECBI Problem, OBVL, DPICS subcategories ‘inappropriate behavior’, ‘child compliance, ‘negative leading’, ‘praise’, and ‘demandingness’). In the PCIT-Clinic sample, mind-mindedness was not significantly correlated to any of the outcome variables (ECBI Intensity, ECBI Problem, PSI-SF, ASR, and all the DPICS subcategories).

We also checked whether mind-mindedness was associated with mothers’ and child characteristics. In the PCIT-Home sample, mind-mindedness was significantly positively correlated with mothers’ educational level, *r*(17) = 0.47, *p* = 0.048, to family income, *r*(18) = 0.56, *p* = 0.012, and to relationship of the mother to the child, with foster mothers showing higher levels of mind-mindedness than biological mothers, *r*(18) = 0.49, *p* = 0.033. We found no significant correlations of mind-mindedness with mothers’ or child characteristics in the PCIT-Clinic sample.

### 5.2. Moderating and Group Effects of Mind-Mindedness

We examined the effect of parental level of mind-mindedness before the intervention (*high* versus *low*) as between factors in the outcome measures of the PCIT intervention, by performing a series of repeated measures ANOVA, with the PCIT intervention (pre-, post- and follow-up) as within factors. Table 2 and Table 3 present the repeated measures ANOVA results: *F*-values for simple, group, and interaction effects with related effect sizes (partial eta squared). The median of mother’s percentage of appropriate mind-mindedness was 3.1 for the PCIT-Home and 2.3 for the PCIT-Clinic sample. The descriptive statistics are presented in the Appendix A, Table A1 and Table A2.

#### 5.2.1. PCIT-Home Sample

Table 2 shows the results of the repeated measures ANOVA for the PCIT-Home sample. A significant intervention effect of PCIT-Home was found for all outcome variables, except for the DPICS child subcategories ‘inappropriate behavior’ and ‘non-compliance’.

We found significant group effects of mind-mindedness for ECBI Problem, *F* = 4.81, *η_p_*^2^ = 0.22, *p* = 0.043, OBVL, *F* = 4.56, *η_p_*^2^ = 0.21, *p* = 0.048, the DPICS child subcategories ‘inappropriate behavior’, *F* = 6.62, *η_p_*^2^ = 0.28, *p* = 0.020 and ‘non-compliance’, *F* = 10.47, *η_p_*^2^ = 0.38, *p* = 0.005, and for the DPICS parental subcategory ‘praise’, *F* = 8.43, *η_p_*^2^ = 0.33, *p* = 0.010 (see Figure 1). Compared to mothers with a medium-low level of mind-mindedness, mothers with a medium-high level of mind-mindedness reported lower scores on the ECBI Problem and OBVL, expressed higher levels of ‘praise’, and their children showed less ‘inappropriate behavior’ and ‘non-compliance’ on the DPICS at pre-, post-, and follow-up measurements. The group effects for ECBI Intensity and the DPICS parental subcategories ‘positive following’, ‘negative leading’, and ‘demandingness’ were not significant (see Table 2 for *F*, *η_p_*^2^, *p* values).

A significant interaction effect of mind-mindedness was found for the DPICS parent subcategory ‘demandingness’, *F =* 3.82, *η_p_*^2^ = 0.18, *p* = 0.032 (see Figure 1). Adjusted SIDAK comparisons showed that mothers with a medium-low level of mind-mindedness displayed significant less ‘demandingness’ at post-measurement than before the intervention (*M_pre_* = 30.90, *SE* = 5.18, *M_post_* = 13.30, *SE* = 1.80), *p* = 0.002. The interaction effects for the other variables (ECBI Intensity, ECBI Problem, OBVL, DPICS child subcategories ‘inappropriate behavior’ and ‘non-compliance’, DPICS parental subcategories ‘positive following’, ‘negative leading’, and ‘praise’) were not significant (see Table 2 for *F*, *η_p_*^2^, *p* values).

#### 5.2.2. PCIT-Clinic Sample

Table 3 presents the results of the repeated measures ANOVA for the PCIT-Clinic sample. A significant intervention effect of PCIT-Clinic was found for all outcome variables. 

Looking at the quality of the parent−child interaction measured with the Dyadic Parent–Child Interaction Coding System (DPICS), a significant interaction effect of mind-mindedness was found for the parental subcategory ‘negative leading’, *F* = 5.40, *η_p_*^2^ = 0.19, *p* = 0.008 (see Figure 2). Adjusted SIDAK comparisons show that mothers with a medium-high level of mind-mindedness, displayed significant more ‘negative leading’ before the intervention than at post-measurement (*M_pre_* = 0.46, *SE* = 0.05, *M_post_* = 0.19, *SE* = 0.03), *p* < 0.001, and at follow-up measurement (*M_follow-up_* = 0.17, *SE* = 0.04), *p* < 0.001. No significant interaction effects of mind-mindedness were found for the child’s behavioral problems (ECBI Intensity, ECBI Problem), parenting stress (PSI-SF) and mother’s anxious and depressive symptoms (ASR). 

Furthermore, no significant group effects of mind-mindedness were found (see Table 3 for *F*, *η_p_*^2^, *p* values).

### 5.3. Premature Attrition and Total Sessions

In the PCIT-Clinic sample, treatment continues until the mastery criteria for successful treatment are met, resulting in a varying number of sessions between families that complete treatment. In the PCIT-Home sample, all families completing treatment received eight sessions in total according to treatment procedure. Accordingly, we performed analyses on associations between level of mind-mindedness and total number of sessions for the PCIT-Clinic sample only. Analyses on associations between level of mind-mindedness and premature attrition were performed for both samples.

#### 5.3.1. PCIT-Home Sample

The PCIT-Home sample (*n* = 19) contained only three dropouts, which were all of mothers with a medium-high level of mind-mindedness (MM). Among the completers (*n* = 16), 37.5% were MM-*high* and 62.5% MM-*low*. A chi-square test of independence, showed mind-mindedness was significantly associated with dropouts, *X*^2^ (1, *n* = 19) = 4.0, *φ* = 0.46, *p* = 0.047.

#### 5.3.2. PCIT-Clinic Sample

We investigated if the level of mind-mindedness (MM) was correlated with treatment dropouts, using the PCIT-Clinic sample which included both treatment completers and dropouts (*n* = 37). Mind-mindedness was distributed differently between treatment completers (*n* = 25; MM-*high* = 40.0%, MM-*low* = 60.0%;) and dropouts (*n* = 12; MM-*high* = 66.7%, MM-*low* = 33.3%). However, a chi-square test of independence showed that there was no significant association between mind-mindedness and dropouts, *X*^2^ (1, *n* = 37) = 2.3, *φ* = 0.25, *p* = 0.129.

To examine whether level of mind-mindedness had an effect on the total number of sessions for a successful treatment, we performed analyses with completers only (*n* = 25). The number of total sessions did not differ significantly between the two mind-mindedness groups (MM-*high M* = 19.39, MM-*low M* = 19.27), *F*(24) = 0.57, *p* = 0.974.

### 5.4. Sensitivity Analyses

For the PCIT-Home sample we performed sensitivity analyses with the educational level of the mother and the relationship of the mother to the child (biological/foster) as covariates to examine the robustness of our finding (see Appendix A, Table A3). We did not add family income as a covariate because it was already significantly correlated with educational level of the mother, *r*(17) = 0.63, *p* = 0.005. Due to missing data of the educational level of the mothers, the total sample for the sensitivity analyses was 18. Results of the sensitivity analyses were largely similar compared to the initial analyses, only the group effect for ECBI Problem and OBVL became non-significant. We did not include sensitivity analyses for the PCIT-Clinic sample, because for this sample mind-mindedness was not significantly correlated with the mothers’ or child’s characteristics or with any of the outcome measures.

## 6. Discussion

This study investigated the effect of parental mind-mindedness measured before the start of the intervention on the outcome of both parent−child interaction therapy provided in a time-limited, home-based setting (PCIT-Home) and in a clinic-based setting (PCIT-Clinic). We expected mind-mindedness to have a moderating and group effect on the outcome measures, with better treatment results for mothers with a medium-high level of mind-mindedness. Furthermore, we investigated whether mind-mindedness was associated with a lower risk of premature attrition and—for the PCIT-Clinic sample only—with fewer sessions.

### 6.1. Moderating Effects

We found a large moderating effect for the parental DPICS subcategory ‘negative leading’ in the PCIT-Clinic sample. Mothers with a medium-high level of mind-mindedness showed a significantly larger decrease in the use of ‘negative leading’ at post and follow-up measurements. The DPICS subcategory ‘negative leading’ consists of expressed commands (e.g., ‘give me that toy’), questions (‘what are you building?’) and negative talk (e.g., criticizing the child) coded during child-led play (CLP). CLP represents the first phase of treatment where the focus is on improving the parent−child relationship and the parent is instructed to follow the child in play, with the negation of commands, questions, and negative talk supporting this process [74]. Although not significant in the primary analysis (*p* = 0.053), but significant in the sensitivity analysis (*p* = 0.044), this pattern was also seen for the DPICS subcategory ‘praise’ in the PCIT-Home sample: mothers with a medium-high level of mind-mindedness improved more in making compliments about their child’s behavior or about their child in general than mothers with a medium-low level of mind-mindedness at post-test and follow-up measurements. Again, supporting a positive parent−child relationship through positive interactions and responding appropriately (e.g., with praise, rather than hostility or ignoring). So, although our findings on the moderating effects of mind-mindedness highlight two different parenting skills in the two PCIT samples, they point toward the same direction; namely, that mothers with a medium-high level of mind-mindedness show more improvement in parenting skills that benefit a positive parent−child interaction. The moderating effect for the DPICS subcategory ‘demandingness’ in the PCIT-Home sample was the exception, because here mothers with a medium-low level of mind-mindedness showed significantly more decrease in the use of ‘demandingness’ post-treatment. However, this effect was lost at follow-up, 2 months after treatment, whereby the difference with mothers with a medium-high level of mind-mindedness was not significant anymore.

### 6.2. Group Effects

Observations of the parent−child interaction in the PCIT-Home sample also showed that children of mothers with a medium-high level of mind-mindedness displayed significantly less inappropriate and non-compliant behavior compared to children of mothers with a medium-low level of mind-mindedness at pre-test, post-test, and follow-up measurements. These results are in line with emerging evidence that mind-mindedness protects against the development of child behavioral problems [44,45,46,47,48]. These mothers with a medium-high level of mind-mindedness also reported their child’s behavior to be less problematic, and they made more use of compliments in interaction with their child. They also reported less parenting stress, with non-clinical scores on the OBVL at post-test and follow-up measurements, while the parenting stress of mothers with a medium-low level of mind-mindedness remained in the clinical range at post-test and follow-up measurements. We interpret these findings as a result of mothers with a medium-high level of mind-mindedness being more orientated to the motives and feelings that underlie the child’s behavior, and therefore being less inclined to experience and label these behaviors as difficult or irritating. Studies show that higher mind-mindedness is associated with less parenting stress [48,75,76], which in turn is associated with less hostility towards the child, explaining why it is easier for parents to express compliments in interaction with their child. Although the other outcome variables (ECBI Intensity, DPICS subcategories ‘positive following’, ‘negative leading’, and ‘demandingness’) were not significant, they did follow the same pattern, with mothers with a medium-high level of mind-mindedness showing better results than mothers with a medium-low level of mind-mindedness at pre-test, post-test (except for DPICS ‘demandingness’) and follow-up assessments.

Nevertheless, we did not find any group effects of mind-mindedness in the PCIT-Clinic sample. An explanation might be the difference in group characteristics, with previous studies also finding that different populations of caregivers display different levels of mind-mindedness [77,78]. The PCIT-Home sample was more heterogeneous, with 32% foster mothers and more higher educated mothers, but after controlling for confounding in the PCIT-Home sample, most of the group effects of mind-mindedness remained, with mothers with a medium-high level of mind-mindedness showing better scores on most outcome measures at pre-test, post-test, and follow-up measurements. A limitation is that we could not control for first-time mothers, because this information was not collected in the PCIT-Home sample. (However, in the PCIT-Clinic sample, first-time motherhood was not significantly correlated with mind-mindedness.) Further research with a larger sample size and more statistical power is needed to determine whether confounding by group characteristics can be excluded entirely, since our results regarding group effects of mind-mindedness are incongruent between the two samples. The results in our PCIT-Home sample do, however, prompt the question of whether adding an intervention that improves parental mind-mindedness before or during PCIT for mothers with a medium-low level of mind-mindedness would add to the efficacy of PCIT with better outcomes for this group.

### 6.3. Premature Attrition and Total Sessions

Our findings on the association between mind-mindedness and premature attrition from treatment were not significant in the PCIT-Clinic sample. In the PCIT-Home sample, the association was significant, but the dropout group was too small (*n* = 3) to draw significant conclusions from. Although against our expectations, because we expected mothers with a medium-high level of mind-mindedness to stay more committed to treatment, these results are in line with Zimmer-Gembeck et al. [79], who also did not find significant differences in parental mentalization for those who did and did not complete PCIT in pre-assessment measures.

For the discussion of our findings, it is relevant to mention that the treatment procedure in the PCIT-C sample complicates measuring effects of mind-mindedness on the outcome variables. Where PCIT-Home has a limited number of eight sessions, PCIT-Clinic continues until the specific criteria for successful treatment are met (total sessions *M* = 19.3, *Range* = 10–39), so it is expected that all families in the PCIT-Clinic roughly have the same scores on the outcome variables at post-test and follow-up measurements. So instead, we expected to find a different distribution in the number of sessions between mothers with a medium-high and medium-low level of mind-mindedness in the PCIT-Clinic sample. However, in line with the less-is-more principle of Bakermans−Kranenburg et al. [32], which indicates that more is not always better, we found no significant difference. In other words, mothers with a medium-low level of mind-mindedness did not need more sessions than mothers with a medium-high level of mind-mindedness to reach the same outcomes in the PCIT-Clinic sample.

### 6.4. Limitations

The present study contained several limitations. First, although the power of our study was roughly large enough, the small sample sizes make our conclusions less robust. Therefore, we did not analyze mind-mindedness as a continuum but as a dichotomous variable. A second possible limitation is the children’s age in the samples, which was between 2 and 7 years. The observation-based measure of mind-mindedness is validated for infancy [39]. The main reason is that children beyond infancy become able to communicate their thoughts and feelings verbally, so parents are less challenged in interpreting their child’s behaviors in terms of mental states. Nevertheless, there are multiple studies providing evidence to measure mind-mindedness with observations after infancy [57,80,81,82]. Moreover, parental mind-mindedness is found to be stable in mothers during the preschool years [81], and was not correlated to the children’s age in our preliminary analyses. A third limitation is that only mothers were included, while research suggests that both parents’ use of mind-mindedness plays an important role in the child’s development and behavior problems [83].

### 6.5. Recommendations for Future Research

Future research should aim at larger groups to increase statistical power and generalizability of findings, and investigate whether mind-mindedness itself is improved by PCIT. To investigate the relationship with other constructs of the parent−child relationship, the observational measurements of mind-mindedness could be complemented with self-report (representational) instruments of parental mentalization or observational instruments that measure parents’ sensitivity. Lastly, both parents should be included in future studies.

## 7. Conclusions

In spite of the limitations, this study shows promising results on the effects of parental mind-mindedness in the treatment of children with behavioral problems with PCIT. Our findings suggest that a higher level of parental mind-mindedness leads to more improvement in parenting skills, encouraging a positive parent−child interaction, a key element of PCIT. Furthermore, we found that, but only for the PCIT-Home sample, parents with a higher level of mind-mindedness showed better results on most outcome measures. Both findings suggest that adding a mind-mindedness-improving intervention prior or during PCIT could benefit mothers with a medium to low level of mind-mindedness.

## Figures and Tables

**Figure 1 ijerph-19-04533-f001:**
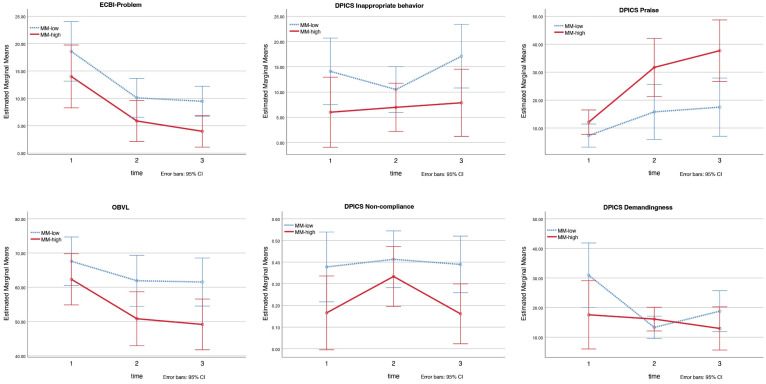
PCIT-Home: group effects for ECBI Problem, OBVL, DPICS inappropriate behavior, non-compliance, praise, and interaction effect for DPICS demandingness. *Note.* MM-low = mothers with a medium-low level of mind-mindedness; MM-high = mothers with a medium-high level of mind-mindedness.

**Figure 2 ijerph-19-04533-f002:**
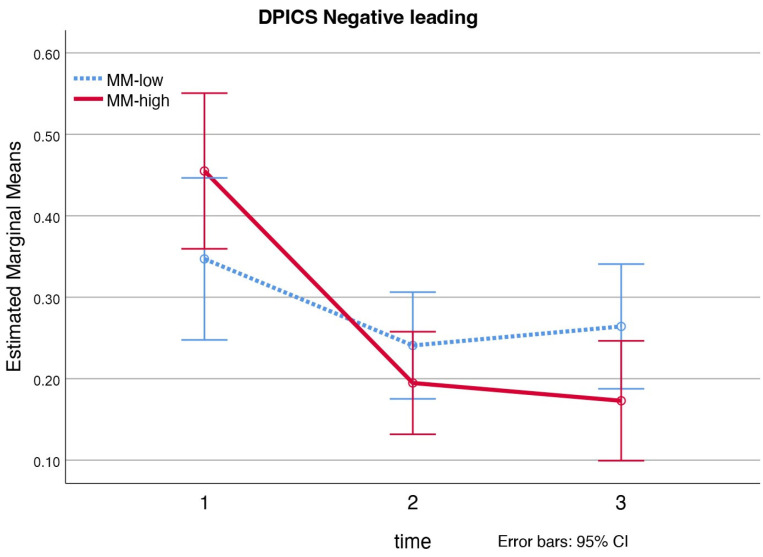
PCIT-Clinic: interaction effect for DPICS negative leading. *Note.* MM-low = mothers with a medium-low level of mind-mindedness; MM-high = mothers with a medium-high level of mind-mindedness.

**Table 1 ijerph-19-04533-t001:** Demographic information for the PCIT-Home and PCIT-Clinic samples.

	PCIT-Home (*n* = 19)			PCIT-Clinic (*n* = 25)		
	*n* (%*)	*M*	*SD*	*n* (%*)	*M*	*SD*
**Child characteristics**						
Age (months)		67.21	19.17		57.88	12.45
Gender (male)	13 (68.4)			18 (72.0)		
Country of birth (Netherlands)	17 (89.5)			24 (96.0)		
Ethnicity (Dutch)	11 (57.9)			20 (80.0)		
Maltreatment history (reported in client file)	14 (73.7)			16 (64.0)		
Conduct problems (SDQ)		4.15	1.82		4.86	2.39
Hyperactivity/inattention problems (SDQ)		7.67	1.78		6.55	2.73
**Family characteristics**						
Age mother (years)		-	-		36.78	6.18
Country of birth mother (Netherlands)	12 (63.2)			20 (80.0)		
Family status (single mother)	8 (42.1)			10 (40.0)		
First-time mothers	-			12 (48%)		
Educational level mother						
*Lower education*	8 (44.4)			13 (52.0)		
*Medium education*	2 (11.1)			2 (8.0)		
*Higher education*	8 (44.4)			6 (24.0)		
Relation to child						
*Biological mother*	13 (68.4)			25 (100)		
*Foster mother*	6 (31.6)			0 (0.0)		
Family income (<€1.000 per month)	1 (5.3)			3 (12.0)		

*Note.* SDQ = Strengths and Difficulties Questionnaire. * Percentage of reported information

**Table 2 ijerph-19-04533-t002:** PCIT-Home repeated measures ANOVA results (*F*-values, *p*, partial eta squared (*η_p_*^2^)).

	Intervention			Group (MM)			Interaction (MM)		
	*F*	*p*	*η_p_* ^2^	*F*	*p*	*η_p_* ^2^	*F*	*p*	*η_p_* ^2^
ECBI Intensity	18.33	0.000	0.52	4.18	0.057	0.20	0.07	0.932	0.00
ECBI Problem	23.05	0.000	0.58	4.81	0.043	0.22	0.09	0.914	0.01
OBVL	16.83	0.000	0.50	4.56	0.048	0.21	2.15	0.132	0.13
DPICS Child									
Inappr. behavior	1.06	0.357	0.06	6.62	0.020	0.28	0.67	0.520	0.04
Non-compliance	1.32	0.280	0.07	10.47	0.005	0.38	0.66	0.521	0.04
DPICS Parent									
Positive following	20.76	0.000	0.55	0.74	0.402	0.04	0.25	0.778	0.02
Negative leading	27.69	0.000	0.62	3.53	0.078	0.17	0.15	0.863	0.01
Praise	17.93	0.000	0.51	8.43	0.010	0.33	3.21	0.053	0.16
Demandingness	6.32	0.005	0.27	1.72	0.207	0.09	3.82	0.032	0.18

*Note.* MM = mind-mindedness.

**Table 3 ijerph-19-04533-t003:** PCIT-Clinic repeated measures ANOVA results (*F*-values, *p*, partial eta squared (*η_p_*^2^)).

	Intervention			Group (MM)			Interaction (MM)		
	*F*	*p*	*η_p_* ^2^	*F*	*p*	*η_p_* ^2^	*F*	*p*	*η_p_* ^2^
ECBI Intensity	23.19	0.000	0.50	2.16	0.155	0.09	0.04	0.965	0.00
ECBI Problem	28.74	0.000	0.56	0.05	0.829	0.00	0.25	0.782	0.01
PSI-SF	6.02	0.005	0.21	0.02	0.880	0.00	1.71	0.193	0.07
ASR Anx./Depr. Scale	6.83	0.003	0.29	0.82	0.376	0.38	0.64	0.534	0.03
DPICS Child									
Inappr. behavior	5.17	0.009	0.18	0.28	0.600	0.02	0.31	0.732	0.01
Non-compliance	4.84	0.012	0.17	0.30	0.592	0.01	0.34	0.717	0.01
DPICS Parent									
Positive following	13.97	0.000	0.38	3.32	0.082	0.13	1.33	0.274	0.06
Negative leading	22.07	0.000	0.49	0.06	0.812	0.00	5.40	0.008	0.19
Praise	6.48	0.003	0.22	0.28	0.599	0.01	1.37	0.265	0.06
Demandingness	22.42	0.000	0.49	0.18	0.678	0.01	2.55	0.089	0.10

*Note.* MM = mind-mindedness.

## Data Availability

Data available on request due to restrictions (e.g., privacy or ethical). The data presented in this study are available on request from the corresponding author. The data are not publicly available due to privacy and ethical guidelines.

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
