# Peer review of "A Cohort Study on the Effect of Parental Mind-Mindedness in Parent−Child Interaction Therapy"

_ijerph, 2022, doi:10.3390/ijerph19084533_

Round 1

Reviewer 1 Report

Overall, it is a very interesting study that underlines the importance of increasing parental mind-mindedness before or during PCIT. I find it really interesting how the authors compare the results between two groups (PCIT-Home and PCIT- Clinic).

Abstract: Please include a sentence including the main conclusion of the study.

Introduction: The authors use the construct Mind-Mindedness and Parental Mentalization interchangeably. Parental mentalization capacity is a broader concept that includes other constructs such as mentality, parental insight and parental reflective functioning. Therefore, in order not to create confusion among readers, I recommend always using the same construct "Mind-Mindedness" (Example: Page 2, lines 86-88).

Procedure: On page 4 (lines 187-189) you indicate that a follow-up was done two months after the post-treatment evaluation for PCIT-Home and six months after the post-treatment evaluation for PCIT-Clinic. What is the reason for this difference in follow-ups?

Measures: Please indicate the internal consistency of both the original instruments and the internal consistency indices you have obtained in the present study.

Participants:

  • The main difference between the sample is that PCIT-HOME includes 31.6% adoptive mothers and PCIT-Clinic contains only birth mothers. Do you think this might be affecting the results?
  • Table 1. Socio-demographic data: Indicate only the % corresponding to the marital status of single mothers. Please report the remaining % corresponding to other marital statuses in the sample.
  • Continuing with the demographic data, please indicate in which employment status these mothers are (active, unemployed, on sick leave...).
  • Have you asked whether they are first-time mothers or not? This information would be important and could affect the level of stress and the development of parenting skills.

Discussion:

It is missing to include a paragraph in relation to future lines of the present study. Some of the future directions of the present study could be the following:

  • The fact that you were able to include observational instruments to assess Mind-Mindedness is a very strong point of your study. This way of assessing Mind-Mindedness could be complemented with a self-report instrument for parents to measure their mentalizing ability.
  • Include fathers in the sample since, as you have indicated, studies underline the important role that both parents' use of mind-mindedness has for the child's development.

Reviewer 2 Report

A Cohort Study on the Effect of Parental Mind-Mindedness in Parent-Child Interaction Therapy (1614477) (Review)

Main message of the article

The paper entitled “A Cohort Study on the Effect of Parental Mind-Mindedness in Parent-Child Interaction Therapy” by Meynen et al examined how parental mind-mindedness – the parent’s propensity to treat the child as an intentional agent – may affect the Parent-Child Interaction Therapy. Generally, the authors reported benefits for mothers with a medium-high level of mind- mindedness in two parenting skills. Also, in the PCIT-Home sample, the authors observed generally better outcomes in mothers with a medium-high level of mind-mindedness.

General Judgment Comments

The paper investigates an important topic: how mind-mindedness in parents may impact parent-child intervention therapy. The main limitations of the current study regard the very small sample sizes and the criteria that were adopted to form the two groups of interest. In fact, both inclusion and exclusion criteria were different in the enrollment for the PCIT-Home and PCIT-Clinic samples. This makes the two groups uncomparable. Although the analysis are appropriate, the statistical power of the results is very little due to the small sample sizes. Also, not always the results are reported in the main text, making the paper hard to follow for the reader. I would also suggest to add some figures in the paper, in order to help the reader visualize the main results.

I would recommend for Major Revision, due to the small sample sizes and for the non-rigorous criteria adopted when making the groups.

Major Issues

  • -  Please clarify the reasons for which the effect of mind-mindedness on attrition rates was studied only for the PCIT- Clinic sample.

  • -  Why did the authors leave foster mothers in the PCIT-Home group? This introduces an important confounding variable in the study and between the two groups. Please clarify.

  • -  How was the presence of disruptive behavioral problems or the risk for child maltreatment assessed to decide about who to enrol in the study? Please clarify.

  • -  The differences between the two samples could only be due to differences in cohorts, since the data collections were conducted in two very different periods of time (the first October 2014 to December 2016 and the second from June 2009 to December 2012). Have the authors controlled for this confounding effect?

  • -  The two groups (PCIT-Home and PCIT-Clinic) were formed by using different inclusion and exclusion criteria, and this choice might add a confounding variable in the results of the current paper. Please explain why different inclusion and exclusion criteria were adopted.

  • -  Line 217-218: “Inter-rater reliability was assessed on 15% of the transcripts”. 15% of reliability does not seem very much. Please clarify how such value was obtained and clarify its implication on the results.

  • -  Please add Cronbach’s Alpha computed for the questionnaires adopted in the current study by taking into account the answers of the used samples.

  • -  What was the adopted statistical software for the analysis?

  • -  Why did the authors used the Multiple Imputation to replace missing data? Please clarify the method and justify its

    adoption.

  • -  Statistical Analysis, line 302: “Most variables were normally distributed”. Please clarify how the authors checked for

    the normality of data and what were the normally distributed variables.

  • -  The authors used the winsorization to correct outliers in the dataset. There are some aspects to clarify. What values

    were considered as outliers? Why did they use the winsorization among other methods to replace outliers?

  • -  Did the authors adopted any correction for the significance levels when conducing multiple testing between variables?

  • -  In the Results section, please report also the non-significant results following the standard format.

  • -  In the Results section report all the significant results in the text too, not only in the tables.

  • -  Generally, it appears that the sample sizes used in the current paper are too small to obtain any reliable conclusion.

    Have the authors conducted a power analysis?

  • -  To help the reader following the paper, I would suggest the authors to add some images/graphs representing the main

    results of their paper.

    Minor Issues

  • -  In the abstract, the authors stated that “A low level of parental mind-mindedness - the parent’s propensity to treat the child as an intentional agent with its own thoughts and emotions – might contribute to premature attrition or cause families to profit less from treatment”. It is not clear whether this is empirical evidence or if, conversely, it is only the hypothesis of the authors. Please clarify.

  • -  Introduction, line 52-53: The sentence “The majority of families that participate in PCIT are families with low resources” seems disconnected from the rest of the text. Please connect it with the text and try to contextualize it. Also, which kind of resources are low in these families (e.g., economical resources, social resources)?

  • -  Introduction, lines 55-57: please justify with a reference the following sentence “Furthermore, other problems such as financial or parental mental health problems might complicate staying motivated for treatment”.

  • -  Introduction, line 62: please justify with a reference the following sentence “while learned skills can be generalized faster to other situations and contexts”.

  • -  Introduction, lines 64-65: “The results on attrition have been inconclusive”. Please specify, inconclusive in which terms?

  • -  Introduction, line 74-76: The transition bringing to the sentence “Little is yet known about whether parental mentalization [33] – the parent’s ability to represent and hold in mind the internal states of their child – affects treatment outcome or attrition” is too abrupt. It is not clear why, after talking about home-based and time-limited adaptations of PCIT, the authors included a paragraph on the parental mentalization. Please clarify in the Introduction, by trying to make the transition to the theme of parental mentalization less abrupt.

  • -  Line 206: “This articles’ first author” to change in “This article’s first author”.

  • -  The section “Sample characteristics” should be moved in the “Participants” section.

    Final comments

    I would recommend for Major Revision.

Round 2

Reviewer 1 Report

The authors have adequately addressed each of the points raised in the review of the article.

Reviewer 2 Report

The article can be accepted. Thank you